# Implicit Maximum a Posteriori Filtering via Adaptive Optimization

**Gianluca M. Bencomo**[1], **Jake C. Snell**[1] **& Thomas L. Griffiths**[1,2]
[1]Department of Computer Science, Princeton University
[2]Department of Psychology, Princeton University
`{gb5435,jsnell,tomg}@princeton.edu`

## Abstract

Bayesian filtering approximates the true underlying behavior of a time-varying system by inverting an explicit generative model to convert noisy measurements into state estimates. This process typically requires either storage, inversion, and multiplication of large matrices or Monte Carlo estimation, neither of which are practical in high-dimensional state spaces such as the weight spaces of artificial neural networks. Here, we frame the standard Bayesian filtering problem as optimization over a time-varying objective. Instead of maintaining matrices for the filtering equations or simulating particles, we specify an optimizer that defines the Bayesian filter *implicitly*. In the linear-Gaussian setting, we show that every Kalman filter has an equivalent formulation using $K$ steps of gradient descent. In the nonlinear setting, our experiments demonstrate that our framework results in filters that are effective, robust, and scalable to high-dimensional systems, comparing well against the standard toolbox of Bayesian filtering solutions. We suggest that it is easier to fine-tune an optimizer than it is to specify the correct filtering equations, making our framework an attractive option for high-dimensional filtering problems.

## 1 Introduction

Time-varying systems are ubiquitous in science, engineering, and machine learning. Consider a multielectrode array receiving raw voltage signals from thousands of neurons during a visual perception task. The goal is to infer some underlying neural state that is not directly observable, such that we can draw connections between neural activity and visual perception, but raw voltage signals are a sparse representation of neural activity that is shrouded in noise. To confound the problem further, the underlying neural state changes throughout time in both expected and unexpected ways. This problem, and most time-varying prediction problems, can be formalized as a probablistic state space model where latent variables evolve over time and emit observations (Simon, 2006). One solution to such a problem is to apply a Bayesian filter, a type of probabilistic model that can infer the values of latent variables from observations.

Typically, Bayesian filters require matrix storage, inversion, and multiplication or the storage of *particles*, which are samples from the filtering distribution at every time step. In large state and observation spaces, the computational costs associated with both of these approaches render them impractical (Raitoharju & Piché, 2019). In addition, since most time-varying systems do not have ground-truth states available, accurately estimating the process noise – the variability of the underlying latent variables – is nearly impossible. This problem is exacerbated in continuous systems where we need to perform numerical integration, which introduces additional uncertainty. Existing Bayesian filters are sensitive to process noise and dynamics misspecification, making them hard to use in these settings (Mehra, 1972). Most filters are highly structured, requiring explicit specification of distributional assumptions. To be usable in practical settings, we need a system that remains effective when scaled to large state spaces (being applicable to weather systems, neural recording, or even modeling the evolving weights of an artificial neural network) and is robust to misspecification of the process noise and dynamics (which is almost guaranteed).

The practical problems that arise when training deep neural networks (DNNs) are not dissimilar from those that make Bayesian filters difficult to work with: high-dimensional state spaces, nonlinearities, and nonconvexity. These pressures have made way for adaptive optimizers (Duchi et al., 2011; Tieleman & Hinton, 2012; Zeiler, 2012; Kingma & Ba, 2015) that offer a largely prescriptive solution for training DNNs, succeeding many previous approaches to optimizing neural networks that include applying Bayesian filtering equations (Haykin, 2004). To efficiently and effectively overcome the difficulties of training DNNs (Glorot & Bengio, 2010), modern optimizers invariably make use of (i) the empirical Fisher information to crudely incorporate curvature at very little computational cost (Martens, 2020) and (ii) momentum to increase stability and the rate of convergence (Sutskever et al., 2013). We show that these same methods can be applied to Bayesian filtering in an extremely practical and theoretically-motivated framework.

In the following sections, we cast Bayesian filtering as optimization over a time-varying objective and show that much of the information that is explicitly defined by structured filters can be *implicitly* internalized by optimizer hyperparameters. In doing so, we can solve many of the scalability challenges associated with Bayesian filters and make Bayesian filtering easier to implement, especially for the well-versed deep learning practitioner. We show that our proposed method, Implicit Maximum a Posteriori (IMAP) Filtering, is robust to misspecification and matches or outperforms classical filters on baseline tasks. We also show that it naturally scales up to high-dimensional problems such as adapting the weights of a convolutional neural network, and that it performs particularly well in this setting. We argue that it is easier to specify an optimizer than it is to correctly identify classical filtering equations, and our results illustrate the benefits of this implicit filtering approach.

## 2 BACKGROUND

We begin with an introduction to the filtering problem. Assume a discrete-time state space model with a Markov process over the states $\mathbf{x}_{0:T} \triangleq \mathbf{x}_0, \mathbf{x}_1, \ldots, \mathbf{x}_T$ and a sequence of observations $\mathbf{y}_{1:T} \triangleq \mathbf{y}_1, \mathbf{y}_2, \ldots, \mathbf{y}_T$ which are conditionally independent given the corresponding state. The joint probability of the states and observations is

$$p(\mathbf{x}_{0:T}, \mathbf{y}_{1:T}) = p(\mathbf{x}_0) \prod_{t=1}^{T} p(\mathbf{x}_t|\mathbf{x}_{t-1}) p(\mathbf{y}_t|\mathbf{x}_t), \tag{1}$$

where $p(\mathbf{x}_0)$ is the *initial distribution*, $p(\mathbf{x}_t|\mathbf{x}_{t-1})$ is the *transition distribution*, and $p(\mathbf{y}_t|\mathbf{x}_t)$ is the *likelihood*. The posterior over the joint state $\mathbf{x}_{1:t}$ of the system given all of the observations up until the current time $\mathbf{y}_{1:t}$ is given by:

$$p(\mathbf{x}_{1:t}|\mathbf{y}_{1:t}) = \int p(\mathbf{x}_0) \prod_{s=1}^{t} \frac{p(\mathbf{x}_s|\mathbf{x}_{s-1}) p(\mathbf{y}_s|\mathbf{x}_s)}{p(\mathbf{y}_s|\mathbf{y}_{1:s-1})} d\mathbf{x}_0, \tag{2}$$

where $p(\mathbf{y}_1|\mathbf{y}_{1:0}) \triangleq p(\mathbf{y}_1)$. In sequential estimation, we are often only interested in the marginal of the current state $p(\mathbf{x}_t|\mathbf{y}_{1:t})$, known as the *filtering distribution*. We normally want to compute or approximate this distribution such that the requisite computational resources do not depend on the length of the sequence. This can be accomplished by first initializing $p(\mathbf{x}_0|\mathbf{y}_{1:0}) \triangleq p(\mathbf{x}_0)$, and then forming a *predictive distribution* via the Chapman-Kolmogorov equation,

$$p(\mathbf{x}_t|\mathbf{y}_{1:t-1}) = \int p(\mathbf{x}_t|\mathbf{x}_{t-1}) p(\mathbf{x}_{t-1}|\mathbf{y}_{1:t-1}) d\mathbf{x}_{t-1}, \tag{3}$$

and updating that predictive distribution, in light of measurements $\mathbf{y}_t$, by a simple application of Bayes' rule for every time step $t = 1, \ldots, T$,

$$p(\mathbf{x}_t|\mathbf{y}_{1:t}) = \frac{p(\mathbf{y}_t|\mathbf{x}_t) p(\mathbf{x}_t|\mathbf{y}_{1:t-1})}{p(\mathbf{y}_t|\mathbf{y}_{1:t-1})}. \tag{4}$$

### 2.1 KALMAN FILTER FOR LINEAR-GAUSSIAN SYSTEMS

When the transition distribution and likelihood are linear and Gaussian, we can compute optimal estimates in closed-form via the Kalman filtering equations (Kalman, 1960). Assume the following

state space model:

$$p(\mathbf{x}_0) = \mathcal{N}(\mathbf{x}_0 \mid \boldsymbol{\mu}_0, \boldsymbol{\Sigma}_0), \tag{5}$$

$$p(\mathbf{x}_t \mid \mathbf{x}_{t-1}) = \mathcal{N}(\mathbf{x}_t \mid \boldsymbol{F}_{t-1}\mathbf{x}_{t-1}, \boldsymbol{Q}_{t-1}), \tag{6}$$

$$p(\mathbf{y}_t \mid \mathbf{x}_t) = \mathcal{N}(\mathbf{y}_t \mid \mathbf{H}_t\mathbf{x}_t, \mathbf{R}_t), \tag{7}$$

where $\boldsymbol{F}_{t-1}$ and $\mathbf{H}_t$ are the transition and observation matrices, respectively, with process noise $\boldsymbol{Q}_{t-1}$ and measurement noise $\mathbf{R}_t$. The Kalman filter propagates the filtering distribution from the previous timestep $p(\mathbf{x}_{t-1} \mid \mathbf{y}_{1:t-1}) = \mathcal{N}(\mathbf{x}_{t-1} \mid \boldsymbol{\mu}_{t-1}, \boldsymbol{\Sigma}_{t-1})$ with the predict step,

$$p(\mathbf{x}_t|\mathbf{y}_{1:t-1}) = \mathcal{N}(\mathbf{x}_t|\boldsymbol{\mu}_t^-, \boldsymbol{\Sigma}_t^-), \text{ where } \boldsymbol{\mu}_t^- = \boldsymbol{F}_{t-1}\boldsymbol{\mu}_{t-1}, \boldsymbol{\Sigma}_t^- = \boldsymbol{F}_{t-1}\boldsymbol{\Sigma}_{t-1}\boldsymbol{F}_{t-1}^\top + \boldsymbol{Q}_{t-1}, \tag{8}$$

and then updates that distribution as follows:

$$p(\mathbf{x}_t|\mathbf{y}_{1:t}) = \mathcal{N}(\mathbf{x}_t \mid \boldsymbol{\mu}_t, \boldsymbol{\Sigma}_t), \text{ where } \boldsymbol{\mu}_t = \boldsymbol{\mu}_t^- + \boldsymbol{K}_t\mathbf{v}_t, \boldsymbol{\Sigma}_t = \left[(\boldsymbol{\Sigma}_t^-)^{-1} + \mathbf{H}_t^\top \boldsymbol{R}_t^{-1}\mathbf{H}_t\right]^{-1}, \tag{9}$$

$\mathbf{v}_t = \mathbf{y}_t - \mathbf{H}_t\boldsymbol{\mu}_t^-$ is the prediction error, and $\boldsymbol{K}_t = \boldsymbol{\Sigma}_t^- \boldsymbol{H}^\top(\mathbf{H}_t\boldsymbol{\Sigma}_t^-\boldsymbol{H}_t^\top + \boldsymbol{R}_t)^{-1}$ is known as the Kalman gain. We discuss further details in Appendix A.1.1.

## 2.2 Approximate Bayesian Filtering for Nonlinear Systems

In real-world systems, guarantees for optimality break down due to inherent nonlinearities and their (at best) approximate Gaussianity. The broader Bayesian filtering community has been motivated by developing methods that can operate in these more realistic settings (Särkkä & Svensson, 2023). Classical solutions include the extended Kalman filter (EKF) and iterated extended Kalman filter (IEKF) that make use of Jacobian matrices to linearize both the dynamics and observation models, and subsequently use the Kalman filtering equations (Gelb, 1974). These are arguably the most popular filtering implementations but only work well in mildly nonlinear systems on short timescales (Julier & Uhlmann, 2004). The unscented Kalman filter (UKF) (Julier et al., 1995; Julier & Uhlmann, 2004), like the particle filter (PF) (Doucet et al., 2001), is better equipped to handle highly nonlinear systems but both of these methods suffer from the curse of dimensionality and sensitivity to misspecified dynamics. Nonetheless, EKFs, IEKFs, UKFs, and PFs are the most prominent filtering solutions found throughout industry and research, leveraging expensive linearization techniques or particle approximations to treat nonlinearities. We provide complete details for these classical approaches to nonlinear filtering in Appendix A.1.

## 3 Bayesian Filtering as Optimization

The update step (4) in a Bayesian filter is a straightforward application of Bayes' rule with the predictive distribution $p(\mathbf{x}_t|\mathbf{y}_{1:t-1})$ playing the role of the prior. In linear-Gaussian systems, this posterior distribution is available in closed-form (9) but one could alternatively achieve an equivalent result via gradient descent due to its connection to regularized least-squares (Santos, 1996). In this section, we describe this connection and how an optimization procedure can implicitly define a prior distribution, a kind of dual formulation of Bayes' theorem (see Figure 1). Such a formulation reduces Bayesian inference to point estimates, but this can be more practical for nonlinear systems whose filtering distributions $p(\mathbf{x}_t|\mathbf{y}_{1:t})$ are not analytically tractable in general.

## 3.1 Finding the Posterior Mode via Truncated Gradient Descent

The mode of the filtering distribution can be expressed as the minimizer of the negative log posterior:

$$\boldsymbol{\mu}_t = \arg\min_{\mathbf{x}_t} \bar{\ell}_t(\mathbf{x}_t), \text{ where } \bar{\ell}_t(\mathbf{x}_t) = -\log p(\mathbf{y}_t|\mathbf{x}_t) - \log p(\mathbf{x}_t|\mathbf{y}_{1:t-1}). \tag{10}$$

In the case of the linear-Gaussian system, the regularized loss function is (up to an additive constant):

$$\bar{\ell}_t(\mathbf{x}_t) = \frac{1}{2}\|\mathbf{y}_t - \mathbf{H}_t\mathbf{x}_t\|_{\mathbf{R}_t}^2 + \frac{1}{2}\|\mathbf{x}_t - \boldsymbol{\mu}_t^-\|_{\boldsymbol{\Sigma}_t^-}^2 ., \tag{11}$$

where $\|\mathbf{z}\|_{\mathbf{P}}^2 \triangleq \mathbf{z}^\top \mathbf{P}^{-1}\mathbf{z}$. Santos (1996) observed that the minimizer of a regularized least squares problem as in (11) can be recovered by performing truncated gradient descent on the likelihood term alone. Specifically, let $\boldsymbol{\mu}_t^{(0)} = \boldsymbol{\mu}_t^-$ and define the recursion

$$\boldsymbol{\mu}_t^{(k+1)} \leftarrow \boldsymbol{\mu}_t^{(k)} + \mathbf{M}_t\mathbf{H}_t^\top \mathbf{R}_t^{-1}(\mathbf{y}_t - \mathbf{H}_t\boldsymbol{\mu}_t^{(k)}), \text{ for } k = 1, \dots, K-1. \tag{12}$$

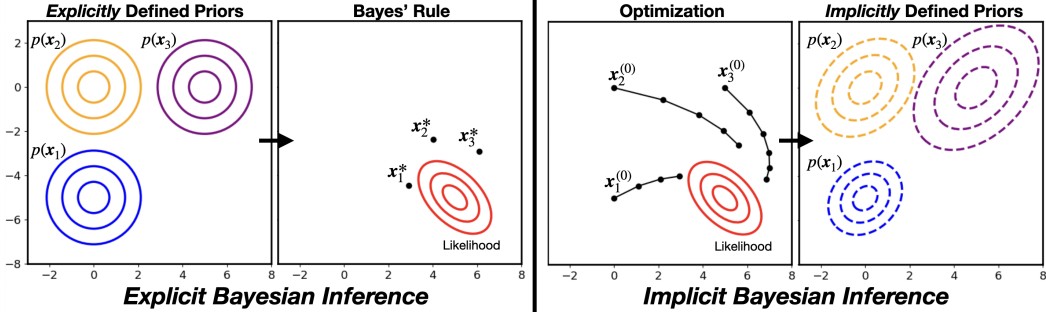

Figure 1: *Explicit Bayesian inference* (**left**) uses Bayes' rule to balance information from the likelihood and explicitly defined priors. $x_i^*$ are MAP estimates of three different posteriors, respective to the priors shown in blue, yellow, and purple. *Implicit Bayesian inference* (**right**) defines an optimization procedure that reverses the directionality of Bayes' rule by defining a procedure to estimate the posterior mode, which corresponds to an implicitly defined prior. Three trajectories $(x_1^{(0)}, \ldots)$, $(x_2^{(0)}, \ldots)$, $(x_3^{(0)}, \ldots)$ of gradient descent are shown with a fixed learning rate over 3, 4, and 5 steps, respectively. Taking the solutions produced by gradient descent as MAP estimates, we can use the corresponding gradient steps to determine the prior distributions that they implied (Santos, 1996).

Let $\mathbf{B}_t$ simultaneously diagonalize $(\boldsymbol{\Sigma}_t^-)^{-1}$ and $\mathbf{H}_t^\top \mathbf{R}_t^{-1} \mathbf{H}_t$ such that $\mathbf{B}_t^\top (\boldsymbol{\Sigma}_t^-)^{-1} \mathbf{B}_t = \mathbf{I}$ and $\mathbf{B}_t^\top \mathbf{H}_t^\top \mathbf{R}_t^{-1} \mathbf{H}_t \mathbf{B}_t = \mathrm{diag}(r_1, \ldots, r_n)$. Such a matrix $\mathbf{B}_t$ is guaranteed to exist since both $(\boldsymbol{\Sigma}_t^-)^{-1}$ and $\mathbf{H}_t^\top \mathbf{R}_t^{-1} \mathbf{H}_t$ are symmetric and $(\boldsymbol{\Sigma}_t^-)^{-1}$ is positive definite (Golub & Van Loan, 2013, Sec. 8.7). The columns of $\mathbf{B}_t$ are the generalized eigenvectors of $\mathbf{H}_t^\top \mathbf{R}_t^{-1} \mathbf{H}_t$ with respect to $(\boldsymbol{\Sigma}_t^-)^{-1}$ and $r_1, \ldots, r_n$ are the corresponding eigenvalues. By Theorem 3.1 of Santos (1996) $\boldsymbol{\mu}_t^{(K)}$ defined in (12) is the minimizer of (11) when $\mathbf{M}_t = \mathbf{B}_t \mathrm{diag}(\lambda_1, \ldots, \lambda_n) \mathbf{B}_t^\top$, where $\lambda_i = (1/r_i)(1 - (1 + r_i)^{-1/K})$ if $r_i \neq 0$ and $\lambda_i = 1$ otherwise. This suggests an optimization wherein $\boldsymbol{\mu}_t$ is computed via $K$ steps of gradient descent on the likelihood, where $\mathbf{M}_t$ is the learning rate matrix induced by $\boldsymbol{\Sigma}_t^-$.

### 3.2 TRUNCATED GRADIENT DESCENT AS PRIOR SPECIFICATION

Instead of computing the learning rate matrix $\mathbf{M}_t$ for a given $\boldsymbol{\Sigma}_t^-$, consider specifying $\mathbf{M}_t$ directly. For example, $\mathbf{M}_t = \rho \mathbf{I}$ recovers gradient descent on the likelihood with learning rate $\rho$. Let $\mathbf{C}_t$ be chosen such that $\mathbf{C}_t^\top \mathbf{M}_t^{-1} \mathbf{C}_t = \mathbf{I}$ and $\mathbf{C}_t^\top \mathbf{H}_t^\top \mathbf{R}_t^{-1} \mathbf{H}_t \mathbf{C}_t = \mathrm{diag}(s_1, \ldots, s_n)$. As for $\mathbf{B}_t$ above, such a $\mathbf{C}_t$ will exist since both $\mathbf{M}_t^{-1}$ and $\mathbf{H}_t^\top \mathbf{R}_t^{-1} \mathbf{H}_t$ are symmetric and $\mathbf{M}_t^{-1}$ is positive definite. Then, by Theorem 3.2 of Santos (1996), $\boldsymbol{\mu}_t^{(K)}$ as defined by (12) is the minimizer of (11) where $\boldsymbol{\Sigma}_t^- = \mathbf{C}_t \mathrm{diag}(\sigma_1, \ldots, \sigma_n) \mathbf{C}_t^\top$, where $\sigma_i = (1/s_i)((1 - s_i)^{-K} - 1)$ if $s_i \neq 0$ and $\sigma_i = 1$ otherwise. Thus, in a linear-Gaussian system, specifying the learning rate matrix $\mathbf{M}_t$ implicitly defines an equivalent predictive covariance $\boldsymbol{\Sigma}_t^-$ and truncated gradient descent on the likelihood recovers the mode of the filtering distribution.

### 3.3 ALTERNATIVE INTERPRETATION AS VARIATIONAL INFERENCE

The procedure shown above can be alternatively motivated by positing a variational distribution $q_t(\mathbf{x}_t) = \mathcal{N}(\mathbf{x}_t \mid \mathbf{m}_t, \mathbf{M}_t)$ with fixed covariance $\mathbf{M}_t$. We show in Appendix A.2 that the truncated optimization of (12) is equivalent to natural gradient ascent on the ELBO. The choice of $\mathbf{M}_t$ then specifies the covariance of the variational approximate to the filtering distribution, the learning rate matrix of the optimization, and the effective prior covariance. More generally, Khan & Rue (2023) demonstrates that many popular adaptive optimization techniques such as RMSprop and Adam can be interpreted as performing variational inference with different choices of $q_t(\mathbf{x}_t)$. In this light, the choice of the optimization algorithm when combined with $k$ steps of optimization implicitly defines a corresponding prior distribution. In the next section, we propose a simple yet effective filtering algorithm based on this interpretation of the update step.

## 4 IMPLICIT MAXIMUM A POSTERIORI FILTERING

In this section, we state the Bayesian filtering steps of our approach, which (partially) optimizes the likelihood $p(\mathbf{y}_t|\mathbf{x}_t)$ at every time step. The filtering distribution at each time $t$ is represented by its mean estimate $\hat{\boldsymbol{\mu}}_t$. In addition to the likelihood, we also assume knowledge of a sequence of transition functions $f_t(\mathbf{x}_{t-1}) \triangleq \mathbb{E}[\mathbf{x}_t|\mathbf{x}_{t-1}]$. This is less restrictive than fully specifying the transition distribution, as done in standard Bayesian filtering. The overall algorithm for the Implicit MAP Filter is summarized in Algorithm 1.

---

**Algorithm 1** Implicit MAP Filter

**Input:** Timesteps $T$, initial state estimate $\hat{\boldsymbol{\mu}}_0$, sequence of loss functions $\ell_1, \ldots, \ell_T$ where $\exp(-\ell_t(\mathbf{x})) \propto p(\mathbf{y}_t \mid \mathbf{x}_t = \mathbf{x})$, sequence of transition functions $f_1, \ldots, f_T$ where $f_t(\mathbf{x}) = \mathbb{E}[\mathbf{x}_t|\mathbf{x}_{t-1} = \mathbf{x}]$, optimizer $\mathcal{M}$, number of optimization steps $K$.
**Output:** Filtering state estimates $\hat{\boldsymbol{\mu}}_1, \ldots, \hat{\boldsymbol{\mu}}_T$.

**for** $t = 1$ **to** $T$ **do**
    $\boldsymbol{\mu}_t^- \leftarrow f_t(\hat{\boldsymbol{\mu}}_{t-1})$
    $\hat{\boldsymbol{\mu}}_t \leftarrow \text{IMAP-UPDATE}(\boldsymbol{\mu}_t^-, \ell_t, \mathcal{M}, K)$
**end for**

---

**Algorithm 2** Update step of Implicit MAP Filter: IMAP-UPDATE($\boldsymbol{\mu}^-, \ell, \mathcal{M}, K$)

**Input:** Initialization $\boldsymbol{\mu}^-$, loss function $\ell$, optimizer $\mathcal{M}$, number of optimization steps $K$.
**Output:** State estimate $\hat{\boldsymbol{\mu}}$.

$\mathbf{m}^{(0)} \leftarrow \boldsymbol{\mu}^-$
**for** $k = 0$ **to** $K - 1$ **do**
    $\mathbf{g}^{(k)} \leftarrow \nabla_\mathbf{x} \ell(\mathbf{x})|_{\mathbf{x} = \mathbf{m}^{(k)}}$
    $\mathbf{m}^{(k+1)} \leftarrow \mathbf{m}^{(k)} + \mathcal{M}(\mathbf{g}^{(k)}; \mathbf{g}^{(0:k-1)})$
**end for**
$\hat{\boldsymbol{\mu}} \leftarrow \mathbf{m}^{(K)}$

---

### 4.1 PREDICT STEP

The predictive distribution for the Implicit MAP Filter is obtained by applying the first-order delta method (Dorfman, 1938; Ver Hoef, 2012) to the Chapman-Kolmogorov equation (3):

$$p(\mathbf{x}_t \mid \mathbf{y}_{1:t-1}) = \int p(\mathbf{x}_t \mid \mathbf{x}_{t-1})p(\mathbf{x}_{t-1} \mid \mathbf{y}_{1:t-1})d\mathbf{x}_{t-1} \approx p(\mathbf{x}_t \mid \mathbf{x}_{t-1} = \boldsymbol{\mu}_{t-1}). \qquad (13)$$

The mean of the predictive distribution is thus obtained by applying the transition function to the previous state estimate: $\mathbb{E}[\mathbf{x}_t|\mathbf{y}_{1:t-1}] \approx f_t(\boldsymbol{\mu}_{t-1})$. This negates the need for matrix-matrix products as is commonly required for the predict step.

### 4.2 UPDATE STEP

The update step is simply the application of optimizer $\mathcal{M}$ for $K$ iterations, considering only an initialization $\boldsymbol{\mu}_t^-$, observation $\mathbf{y}_t$, and loss function $\ell_t$. Here, $\mathcal{M}$ takes as input the history of gradients and outputs a vector to be added to the current state. This formulation captures many popular optimizers, including SGD, RMSProp, and Adam. Our algorithm for the update step of the Implicit MAP Filter is provided in Algorithm 2.

When choosing $\mathcal{M}$ and $K$, the goal is to specify an inductive bias that corresponds to the true filtering equations for the system. Choices such as increasing $K$ and the learning rate increase the Kalman gain (De Freitas et al., 2000). The nature of each optimization step, as well as $K$, defines the shape and entropy of the filtering distribution assumed (Duvenaud et al., 2016). Changing the initialization $\boldsymbol{\mu}_t^-$ given by the predict step modifies the location of the *implicitly* defined prior distribution within the latent space. This concert of initialization $\boldsymbol{\mu}_t^-$, optimizer $\mathcal{M}$, and number of iterations $K$ produces a state estimate $\hat{\boldsymbol{\mu}}_t$ that maximizes the posterior with respect to an *implicitly* defined prior. The aim when specifying these quantities is to define the *correct* posterior.

## 5 EXPERIMENTAL EVALUATION

The goal of our experiments is to assess whether the most popular adaptive optimizers can be used to design effective, robust, and scalable Bayesian filters. We begin with a simple low-dimensional system with nonlinear dynamics to establish that our approach is competitive with standard filters. Then, we turn to a more challenging system that has been used as a benchmark for these algorithms,

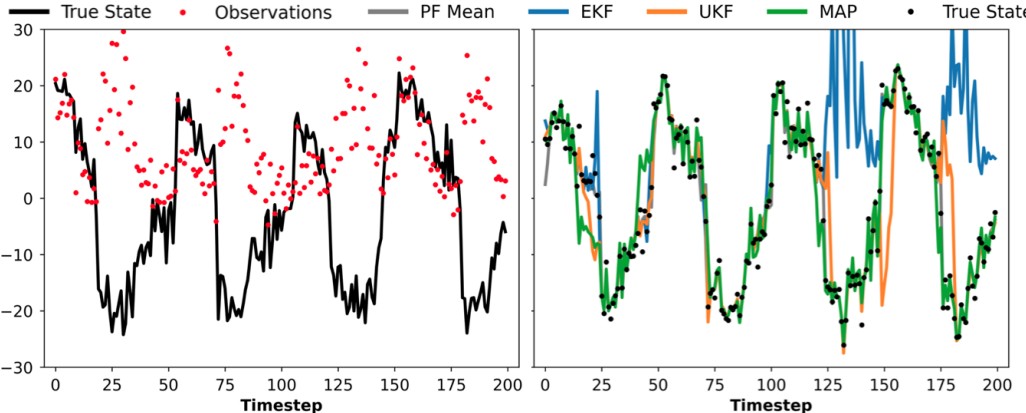

Figure 2: Toy nonlinear system ($\boldsymbol{Q}_{t-1} = 3, \boldsymbol{R}_t = 2$) for a single random trajectory **(left)** and the respective filter estimates **(right)**. Implicit MAP Filter (MAP) shown uses the Adam optimizer with 50 steps at learning rate $\eta = 0.1$ and exponential decay rates $\beta_1, \beta_2 = 0.1$.

demonstrating our advantages. Finally, we show how Implicit MAP Filtering can be effective in high-dimensional settings, such as adapting the weight space of a convolutional neural network.

## 5.1 TOY NONLINEAR SYSTEM

The equivalence between Kalman filtering and $K$ steps of gradient descent elucidated in Section 3.2 suggests that the proposed framework should produce reliable estimates in *approximately* linear-Gaussian systems. To better understand filtering amidst nonlinearites, we first consider the one-dimensional example used in Doucet et al. (2001); Gordon et al. (1993); Kitagawa (1996), originally proposed by Netto et al. (1978), which admits highly nonlinear periodic behavior (see Figure 2):

$$\mathbf{x}_t = \frac{1}{2}\mathbf{x}_{t-1} + 25\frac{\mathbf{x}_{t-1}}{1 + \mathbf{x}_{t-1}^2} + 8\cos\left(1.2t\Delta t\right) + \boldsymbol{q}_{t-1}, \qquad \mathbf{y}_t = \frac{\mathbf{x}_t^2}{20} + \boldsymbol{r}_t, \qquad (14)$$

where $\mathbf{x}_0 \sim \mathcal{N}(0, 1)$, $\boldsymbol{q}_{t-1} \sim \mathcal{N}(0, \boldsymbol{Q}_{t-1})$, $\boldsymbol{r}_t \sim \mathcal{N}(0, \boldsymbol{R}_t)$, and $\Delta t = 0.1$. We simulate ground truth trajectories and their measurements for 200 time steps and evaluate performance using the root mean square errors (RMSEs) with respect to the simulated true trajectories for every combination of $\boldsymbol{Q}_{t-1} \in \{1, 3, 5\}$ and $\boldsymbol{R}_t \in \{1, 2, 3\}$. The means and 95% confidence intervals of the RMSEs are computed from 100 independent Monte Carlo (MC) simulations. For each optimizer, we report the best hyperparameters found by a grid search using 5 separate MC simulations (see Appendix A.4).

Table 1 shows the results for $\boldsymbol{R}_t = 2$ and $\boldsymbol{Q}_{t-1} = 1, 3, 5$. There exist RMSprop and Adam hyperparameters that are competitive with the unscented Kalman filter (UKF) under low-to-medium process noise and better under high process noise. Both the extended Kalman filter (EKF) and iterated extended Kalman filter (IEKF) have the tendency to diverge in this system. The particle filter (PF) performs the best universally, since 1000 particles can capture uni-modal and bi-modal filtering distributions in one dimension quite well. Nonetheless, our Implicit MAP Filter tends to produce estimates that match the modes of the particle filter (see Appendix A.5.1) only struggling on some parts of the sequence with rapid changes in the state. We describe the implementation details for these baseline methods in Appendix A.1 and show similar results for $\boldsymbol{R}_t = 1, 3$ in Appendix A.5.1.

The Implicit MAP Filter does not explicitly use $\boldsymbol{Q}_{t-1}$ or $\boldsymbol{R}_t$ but internalizes both of these terms by the choice of optimizer. In fact, our Implicit MAP Filter can always arbitrarily set $\boldsymbol{R}_t = \boldsymbol{I}$, reducing the objective to the mean squared error loss (see Appendix A.3 for justification). Compared to methods that similarly use gradient information (EKF, IEKF), our Implicit MAP Filters with Adam and RMSprop see a significant performance boost despite the challenge of a non-convex likelihood. We can attribute this boost to the use of momentum (see Appendix A.5.1). In contrast to methods that use sigma points (UKF) or particles (PF), we see competitive performance while avoiding covariance calculations or maintaining many copies of the state.

Table 1: RMSEs on the toy nonlinear system ($R_t = 2$). Results show the average RMSE over 100 MC simulations with 95% confidence intervals.

| Method | RMSE ($Q_{t-1} = 1$) | RMSE ($Q_{t-1} = 3$) | RMSE ($Q_{t-1} = 5$) |
|---|---|---|---|
| EKF | $24.619 \pm 3.988$ | $34.909 \pm 2.732$ | $45.646 \pm 5.629$ |
| IEKF ($K = 5$) | $9.277 \pm 0.652$ | $15.321 \pm 0.493$ | $17.898 \pm 0.559$ |
| IMAP (Adadelta)* | $32.008 \pm 8.650$ | $23.152 \pm 3.973$ | $26.462 \pm 5.058$ |
| IMAP (Gradient Descent)* | $5.589 \pm 0.219$ | $7.966 \pm 0.180$ | $10.130 \pm 0.196$ |
| IMAP (Adagrad)* | $5.569 \pm 0.215$ | $6.549 \pm 0.223$ | $9.264 \pm 0.227$ |
| IMAP (RMSprop)* | $5.304 \pm 0.256$ | $6.000 \pm 0.227$ | $8.527 \pm 0.441$ |
| IMAP (Adam)* | $5.699 \pm 0.190$ | $5.842 \pm 0.231$ | $7.964 \pm 0.334$ |
| UKF | $4.550 \pm 0.242$ | $5.762 \pm 0.270$ | $9.555 \pm 0.400$ |
| PF ($n = 1000$) | $\mathbf{1.565 \pm 0.047}$ | $\mathbf{2.800 \pm 0.108}$ | $\mathbf{4.518 \pm 0.160}$ |

*Methods where the reported hyperparameters were found via grid search (see Appendix A.4).

## 5.2 STOCHASTIC LORENZ ATTRACTOR

In the previous section, we show that our approach is competitive with a particle filter (PF) and the unscented Kalman filter (UKF) in a system that is relatively easy for these methods. Now, we evaluate a mildly nonlinear chaotic system that is more complicated because the stochastic differential equation (SDE) is unreliable for accurately predicting the next state (Evensen, 2009).

Consider a stochastic Lorenz '63 system (Lorenz, 1963; Li et al., 2020; Zhao & Särkkä, 2021),

$$\mathrm{d} \begin{pmatrix} x_{1,t} \\ x_{2,t} \\ x_{3,t} \end{pmatrix} = \begin{pmatrix} \sigma(x_{2,t} - x_{1,t}) \\ x_{1,t}(\rho - x_{3,t}) - x_{2,t} \\ x_{1,t} \cdot x_{2,t} - \beta x_{3,t} \end{pmatrix} \mathrm{d}t + \alpha \mathrm{d}W_t, \qquad \mathbf{y}_t = \begin{pmatrix} x_{1,t} \\ x_{2,t} \\ x_{3,t} \end{pmatrix} + \boldsymbol{r}_t, \qquad (15)$$

where $\mathbf{x}_0 \sim \mathcal{N}(\boldsymbol{\mu}_0, \boldsymbol{I}_3)$ for $\boldsymbol{\mu}_0 = (10, 10, 10)$, $\boldsymbol{r}_t \sim \mathcal{N}(\mathbf{0}, 2 \cdot \boldsymbol{I}_3)$, $\sigma = 10$, $\rho = 28$, $\beta = \frac{8}{3}$, $\alpha = 10$, and $W_t$ is a three-dimensional Wiener process with unit spectral density. Similar to Zhao & Särkkä (2021), we simulate numerical ground truth trajectories and their measurements uniformly for 200 time steps with $\mathrm{d}t = 0.02$ using 10000 Euler-Maruyama integration steps between each measurement. We compute RMSEs and 95% confidence intervals from 100 independent MC simulations and report the best hyperparameters found by a separate grid search (see Appendix A.4).

We report three experimental configurations where we only vary the method used for numerical integration: 4th order Runge-Kutta (RK4) method, Euler's method (Euler), and no integration, assuming a Gaussian random walk (GRW). These methods are presented in decreasing order of accuracy with respect to the aforementioned Euler-Maruyama integration steps. The objective is to assess robustness to dynamics misspecification, which is an unavoidable reality wherever numerical integration is required. Unlike the toy nonlinear system, where the true process noise covariance is known exactly, numerical integration introduces additional unmodeled noise to the process. This presents a realistic challenge that can significantly distort estimates (Huang et al., 2017; Mehra, 1972).

Table 2 shows the results of these experiments and the number of values stored to describe the state ($N$). For fair comparison, we show the EKF and UKF with $Q_{t-1}$ optimized by grid search (see Appendix A.4.1) in addition to the EKF and UKF with the $Q_{t-1}$ specified by the system. Performing such a search for the PF is impractical due to the cost of numerically integrating each particle. The Implicit MAP Filter with gradient descent and the EKF with optimized $Q_{t-1}$ show the best performance. Both are relatively robust to worsening estimates of the SDE. The EKF with optimized $Q_{t-1}$ is close to optimal due to the convexity of this system.

The performance of the Implicit MAP Filter and the optimized EKF are statistically equivalent but with several key differences. Our approach only computes a point estimate over a minimal state space representation, which decreases space complexity and avoids expensive matrix-matrix computations. Classical filters such as the EKF are difficult to tune (Chen et al., 2018) whereas simple adaptive optimizers, like Adam, are fixed to 3 or 4 hyperparameters regardless of the state space and are faster to evaluate, making search quick and effective. The EKF is not well suited for nonconvexities, as shown in Section 5.1, whereas the Implicit MAP Filter overcomes this pitfall by its choice of optimizer.

Table 2: RMSEs on the stochastic Lorenz attractor ($\alpha = 10$). Results show the average RMSE over 100 MC simulations with 95% confidence intervals. RK4 indicates 4th order Runge-Kutta method. Euler indicates Euler's method. GRW indicates a Gaussian random walk. $N$ is the number of values stored to describe the state space ($D = 3$).

| Method | RMSE (RK4) | RMSE (Euler) | RMSE (GRW) | $N$ |
|---|---|---|---|---|
| EKF (original $\boldsymbol{Q}_{t-1}$) | $0.890 \pm 0.094$ | $2.308 \pm 0.155$ | $3.057 \pm 0.037$ | $D + D^2$ |
| EKF (optimized $\boldsymbol{Q}_{t-1}$)* | $\mathbf{0.692 \pm 0.014}$ | $\mathbf{0.952 \pm 0.011}$ | $\mathbf{1.561 \pm 0.010}$ | $D + D^2$ |
| IEKF ($K = 5$) | $0.890 \pm 0.094$ | $2.308 \pm 0.158$ | $3.057 \pm 0.038$ | $D + D^2$ |
| IMAP (Adadelta)* | $6.034 \pm 0.634$ | $7.649 \pm 0.254$ | $10.754 \pm 0.279$ | $\boldsymbol{D}$ |
| IMAP (Adam)* | $0.982 \pm 0.135$ | $1.153 \pm 0.014$ | $1.885 \pm 0.010$ | $\boldsymbol{D}$ |
| IMAP (Adagrad)* | $0.907 \pm 0.144$ | $1.096 \pm 0.017$ | $1.765 \pm 0.013$ | $\boldsymbol{D}$ |
| IMAP (RMSprop)* | $0.881 \pm 0.114$ | $1.081 \pm 0.015$ | $1.757 \pm 0.015$ | $\boldsymbol{D}$ |
| IMAP (Gradient Descent)* | $\mathbf{0.701 \pm 0.018}$ | $\mathbf{0.960 \pm 0.012}$ | $\mathbf{1.561 \pm 0.010}$ | $\boldsymbol{D}$ |
| UKF (original $\boldsymbol{Q}_{t-1}$) | $2.742 \pm 0.059$ | $2.856 \pm 0.066$ | $5.628 \pm 0.067$ | $D + D^2$ |
| UKF (optimized $\boldsymbol{Q}_{t-1}$)* | $1.402 \pm 0.010$ | $1.417 \pm 0.011$ | $1.736 \pm 0.013$ | $D + D^2$ |
| PF ($n = 1000$) | $1.568 \pm 0.027$ | $1.725 \pm 0.031$ | $14.346 \pm 0.365$ | $nD$ |

*Methods where the reported hyperparameters were found via grid search (see Appendix A.4).

Accommodating worsening dynamics estimates is straightforward by either increasing the number of steps or learning rate, each of which corresponds to a weaker prior. In Table 2, our Implicit MAP Filter with gradient descent uses 3 gradient steps with learning rate $\eta = 0.05$ with RK4, 3 steps with $\eta = 0.1$ for the Euler case, and 10 steps with $\eta = 0.1$ for the GRW case. However, the same optimizer settings generally show robustness from RK4 to Euler (see Appendix A.5.2). When visualizing trajectories (see Appendix A.5.2), our approach does a remarkable job of fitting this system despite every initialization producing a vastly different filtering problem. We report additional results for this system using $\alpha = 1, 5, 20$ in Appendix A.5.2.

## 5.3 YEARBOOK

Suppose we have a pre-trained neural network that performs inference in a time-varying environment. This could be a recommender system that must evolve with changing seasons or a facial recognition system that must be robust to temporal shift. Let us assume that data curation is expensive, so we are only given a small number of examples at every time step for adaptation. We can formalize this as the following state space model (Haykin, 2004):

$$\mathbf{w}_t = f(\mathbf{x}_{t-1}) + \boldsymbol{q}_{t-1}, \qquad \mathbf{y}_t = h(\mathbf{w}_t, \boldsymbol{X}_t) + \boldsymbol{r}_t \tag{16}$$

where $\mathbf{w}_t$ is the network's *ideal* weight parameterization, $\boldsymbol{q}_{t-1}$ is some noise vector that corrupts estimates from the transition function $f$, $\mathbf{y}_t$ is the network's desired response vector after applying the transformation $h$ parameterized by $\mathbf{w}_t$ with feature matrix $\boldsymbol{X}_t$, and $\boldsymbol{r}_t$ is the measurement noise.

In this final experiment, we design a time-varying prediction task as described above using the Yearbook dataset (Ginosar et al., 2015), which contains 37,921 frontal-facing American high school yearbook photos that capture changes in fashion and population demographics from 1905 - 2013. We use the data from 1905-1930 to pre-train a 4 layer convolutional neural network (CNN) to perform a binary gender classification task. From 1931-2010, we adapt this weight initialization at each year sequentially using five simple approaches: static weights, direct fit, variational Kalman filtering (VKF), particle filtering (PF), and our Implicit MAP filtering. 32 randomly chosen training examples are available at every time step and a held-out test set of roughly 100 images per time step is used validate the approaches (see Appendix A.5.3 for complete details).

This system is complicated by (1) a state space of 28,193 parameters, (2) unknown weight-space dynamics equations, and (3) unknown process noise and measurement noise covariances. To understand our capacity for modeling such a system given a VKF, PF, and Implicit MAP Filter, we test 5 hyperparameter configurations of each method (see Appendix A.5.3). We report classification accuracies for all five methods in Table 3, grouped into two 40 year blocks. Static weights yields the worst performance due to the problem being fundamentally non-stationary. On the other extreme, attempting to exactly fit the data at each time step severely overfits, which proves to be better but

Table 3: Classification accuracy with $95\%$ confidence intervals over two 40 year blocks in the yearbook dataset.

| Method | % Correct (Validation Years) | % Correct (Test Years) |
|---|---|---|
| Static Weights | $82.416 \pm 2.270$ | $60.897 \pm 1.615$ |
| Particle Filter | $86.430 \pm 1.790$ | $66.715 \pm 2.390$ |
| Variational Kalman Filter | $93.087 \pm 1.038$ | $79.967 \pm 2.204$ |
| Direct Fit | $94.416 \pm 0.924$ | $80.449 \pm 1.845$ |
| Implicit MAP Filter | $\mathbf{94.973 \pm 0.837}$ | $\mathbf{84.747 \pm 2.030}$ |

still suboptimal. The VKF, PF, and Implicit MAP Filter attempt to internalize the system's true state space equations via a small grid search, but this is clearly difficult to do exactly in such a state space. Our implicit approach is less sensitive to this inevitable misspecification. The Implicit MAP Filter with 50 steps of Adam optimization not only showed the best performance, but also highly desirable space and time complexity. In cases where we do not start with a pre-trained network, we can simply fit the first timestep to convergence, which is equivalent to an uninformative prior on the initial state.

## 6 RELATED WORK

Bayesian filtering, optimization, and neural networks share a long history that dates back to early attempts of using the extended Kalman filter (EKF) algorithm to train multilayer perceptrons (Singhal & Wu, 1988). Since then, Kalman filter theory for the training and use of neural networks saw rich development (Haykin, 2004), but fell out of fashion as gradient descent proved to be the more efficient and scalable option for the training of deep neural networks (Bengio et al., 2017). Ruck et al. (1992) realized a connection between optimization and Bayesian filtering, showing that the standard backpropagation algorithm can be seen as a degenerate form of the EKF. Several works have since connected the EKF and the Gauss-Newton method (Bertsekas, 1996; Bell & Cathey, 1993) and showed that the EKF can be seen as a single iteration of Newton's method for a specific quadratic form (Humpherys et al., 2012). Ollivier (2018; 2019) proved equivalence between online natural gradient descent and the EKF when the dynamics are stationary or the process noise is proportional to the posterior covariance over states. Aitchison (2020) used the Bayesian filtering formalism to derive adaptive optimizers like Adam and RMSprop. De Freitas et al. (2000) showed that the process noise and measurement noise can be viewed as adaptive per-parameter learning rates. In the present work, we establish similar connections between optimization and Bayesian filtering but focus on optimization as a tool for Bayesian filtering rather than Bayesian filtering as a tool for optimization.

Several works have similarly used popular optimization methods to address challenges with classical Bayesian filters. Auvinen et al. (2010) and Bardsley et al. (2013) used limited memory Broyden–Fletcher–Goldfarb–Shanno (LBFGS) and conjugate gradients, respectively, to derive variational Kalman filters with low storage covariance, and inverse covariance, approximations. These methods scale up classical filters, like low-rank extended Kalman filters Chang et al. (2023), rather than providing a general filtering approach that is painless to implement. Chen (2003) describes particle filters that move every particle down their respective gradient prior to sampling from the proposal distribution. This is reminiscent of our approach in that it uses gradients, but differs in that it maintains particles, performs sampling, and again falls within the class of explicit filters.

## 7 CONCLUSION

We have shown that Bayesian filtering can be considered as optimization over a time-varying objective and that such a perspective opens the door to effective, robust, and scalable filters built from adaptive optimizers. This framework, however, comes at the cost of interoperability and uncertainty estimates, which limits use in risk-sensitive environments or situations where the filtering equations themselves are used for analysis. Nonetheless, our proposed Implicit MAP Filtering approach is an attractive option for the performance-driven practitioner.

REPRODUCIBILITY STATEMENT

All results and figures reported can be reproduced with the code made available at: github.com/gianlucabencomo/implicitMAP.

ACKNOWLEDGMENTS

We thank R. Thomas McCoy, Jianqiao Zhu, and Logan Nelson and Ryan Adams for helpful discussions. JCS was supported by the Schmidt DataX Fund at Princeton University made possible through a major gift from the Schmidt Futures Foundation. GMB, JCS, and TLG were supported by grant N00014-23-1-2510 from the Office of Naval Research.

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
