# OpenReview forum: "Implicit Maximum a Posteriori Filtering via Adaptive Optimization"
_ICLR.cc/2024/Conference — ICLR 2024 poster_

### Official Review · Reviewer_KcGp · 2023-10-16

**Soundness:** 3 good
**Presentation:** 3 good
**Contribution:** 3 good
**Rating:** 8
**Confidence:** 4

**Summary:**

The paper explains how sequentially optimising the likelihood of a nonlinear time-series estimation problem with any given optimiser can be interpreted as implicitly (point-)estimating the solution of a specific filtering problem.
The difference to existing interpretations of filtering algorithms through the lens of optimisation problems (e.g. the iterated extended Kalman filter) is that the proposed approach only requires access to an optimisation routine, a sequence of likelihood functions, and the mean of a temporal transition model.
*The Kalman filter's need for prescribing process noise covariance functions of the temporal transition model is replaced by the need for prescribing optimisation algorithms.*
Most popular optimisation algorithms scale to high-dimensional settings; most Kalman filter variations do not (because they use covariance information).
This approach takes inspiration from a result by Santos (1996) that states the equivalence between truncated gradient descent and solutions of least-squares-type problems.
The scalability of the resulting algorithm is demonstrated, among other things, by applying the method to a high-dimensional state-space model representing the weight space of a neural network.

**Strengths:**

The paper is well written with a high level of clarity, including the experiment selection, and the algorithm is an enjoyable application of Santos' (1996) result to (nonlinear) filtering problems.
The resulting algorithm is so simple that anyone with previous exposure to optimisation or filtering algorithms could implement it.

I appreciate the honest assessment of limitations; for instance, the method does not deliver any uncertainty quantification.
Overall, I think this paper is in good shape.

**Weaknesses:**

While I do like the paper, I think that the algorithm/presentation has some (minor) issues:

1. **From linear to nonlinear problems:** For linear problems, Santos (1996) shows that a choice of a (truncated) gradient descent corresponds to an implicit choice of predictive covariances.
Thus, for linear problems, estimating the filtering-mean by applying a suitably truncated version of gradient descent "makes sense" -- provided the prior model implicit to the length-scale in gradient descent "makes sense". Using the same idea for nonlinear problems, arbitrary optimisers, and possibly optimisers with momentum is, technically speaking, a leap of faith, and there is nothing to support this transition other than "it seems to work well on the test problems". Fortunately, it does seem to work well on the test problems. (Unless one chooses to use AdaDelta, that is.)

2. **Sensitivity to misspecification:** I am unsure about how, in some parts of the manuscript, the paper describes the implicit MAP filter to be robust against misspecification of the prior. For example, the last paragraph on page 6 reads:

    "Since the Implicit MAP Filter does not explicitly use the process noise $Q_t$, there is no sensitivity to misspecification, as indicated by the empty columns in Table 1."

    I think this statement might be somewhat misphrased because choosing the "wrong" optimiser is not unlike prescribing the wrong process noise covariance.
    In other words, while the implicit MAP filter is, by construction, not sensitive to _explicit_ misspecification of the prior model, it can suffer from misspecification: according to Santos' (1996) results, choosing the wrong length-scale for truncated gradient descent leads to similarly inaccurate approximations as passing the false process noise to a Kalman filter. If we extrapolate a bit by replacing "length-scale" with "optimiser" in the sentences above, we could think of choosing the wrong optimiser as equivalent to selecting the wrong prior model -- this would be model misspecification. The perspective of the wrong optimiser not being unrelated to model-misspecification is corroborated by Table 1: In Table 1, AdaDelta performs *much* worse than other optimisers.


In general,
I think the first point is a clear disadvantage, but a disadvantage one has to live with (unless I have missed something, in which case I am open to correcting my assessment).
The second point is minor because it mainly affects the phrasing in Section 5.1. It should be relatively straightforward to fix.


In general, I think that the strengths outweigh the weaknesses, so I vote for accepting this submission.

**Questions:**

None

---

> ### Author Response · Authors · 2023-11-11
>
> Thank you for your constructive review and honest assessment of our work! We are preparing a revised manuscript that will hopefully address your concerns. Please see our initial responses below.
>
> [Sensitivity to Misspecification]
>
> - You are correct that the wrong optimizer produces the wrong model – same as explicit formulations (EKF, UKF, etc.). However, explicit formulations assume a process noise covariance and measurement noise covariance, a priori, whereas our approach always requires a grid search. The benefit of our approach is that this grid search is not impractical, whereas the increased number of free parameters in explicit formulations makes search more difficult.  Finding the hyperparameters of Adam (learning rate, number of steps, decay terms) is much easier than finding the correct N x N process noise covariance…so we could specify the “correct” model more easily. Further, inference is faster, so search is faster (in addition to the search space being smaller).
>
> [Transition from linear to nonlinear]
>
> - We agree that there is some intuition that is lost when using adaptive optimizers over nonlinear problems. I think we can remedy this issue (in part) by being more clear about how the number of optimizer steps, choice of optimizer, and choice of initialization affect the “implicit” prior. We will adjust section 4.2 to improve clarity.
>
> Thank you again for your feedback, especially your point about loss of intuition in the nonlinear case. We are adjusting accordingly and look forward to discussion!

---

> > ### Comment · Reviewer_KcGp · 2023-11-13
> >
> > Thanks for the reply and for elaborating!
> >
> > To clarify, my assessment regarding "Sensitivity to misspecification" refers to the wording (mostly) in Section 5.1., and not to the advantages/disadvantages of tuning an optimiser vs. tuning a process noise (which is discussed adequately in the submission). I meant that both approaches suffer from model misspecification, even though they manifest slightly differently, and that statements like the one I quoted in my review might mislead readers and would benefit from a correction.
> >
> > Similarly, my assessment regarding "Transition from linear to nonlinear" does not pertain to losing intuition - quite the opposite. The intuition is clear, but my review assesses that the technical foundation must necessarily be abandoned because Santos' results cover only linear problems. I consider this a disadvantage of the approach, albeit one that one has to live with. (As reflected by my overall positive score, I do not think this disadvantage is too grave.).
> >
> > I am happy to see that you will revise the manuscript, and I hope that my clarification helps. What do you think?

---

> ### Author Response · Authors · 2023-11-17
> **Response: New version of the paper uploaded**
>
> Thank you for your response and clarification. We certainly took into consideration your sensitivity to misspecification point and tried to improve the intuition, even if some of it was already there.  We just posted a revised version that we think is a significant improvement. Please see the top-level comment for a list of changes to the paper.
>
> [Response to weaknesses/questions]
> - We think we fully addressed your points regarding clarity and presentation. Thank you for the feedback. We believe that the current version of the paper is a significant improvement to what was originally submitted.
>
> Please let us know if you have any questions or further concerns. It would be great to hear what you think about these adaptations.

---

> > ### Comment · Reviewer_KcGp · 2023-11-17
> >
> > Thank you. I agree with you that the clarity of the presentation in Section 5 has improved and consider all of my questions answered now.

---

### Official Review · Reviewer_zJkz · 2023-11-05

**Soundness:** 3 good
**Presentation:** 4 excellent
**Contribution:** 3 good
**Rating:** 6
**Confidence:** 3

**Summary:**

This work proposes methodology for recursively obtaining point estimates of the states in a Bayesian filtering setting. Specifically, the authors first show that the Kalman filter can be equivalently phrased as a recursive truncated gradient-descent. They then explain that certain hyperparameters used by the optimiser (e.g., the number of gradient steps taken at each time step) can be interpreted as implicitly specifying a particular process-noise variance. Finally, the authors propose to linearise the mean of the state-transition equation in order to deal with models in which this mean is a non-linear function of the previous state.

The authors argue that their methodology is more robust than Kalman- or particle-filtering type methods because it does not need to know true the process-noise variance.

**Strengths:**

**Originality**
Although it relies heavily on previous work by Santos (1996) (as the authors make clear), the proposed approach seems novel.

**Quality**
I believe that this work is of sufficient quality to be, in principle, publishable in ICLR.

**Clarity**
Overall, the presentation is clear. I only have some minor concerns detailed below.

**Significance**
In higher-dimensional problems, in which Monte Carlo sampling may become prohibitively costly, I agree that optimisation-based approaches (like the present work) can be sensible. And the "Yearbook" example seems to support this view. In this regard, I believe that this work makes a worthy contribution.

**Weaknesses:**

**Choice of hyperparameters**
I think Table 1 could slightly misleading because the hyperparameters of the proposed methodology are optimised based on a grid search using (other?) data based on (presumably) the same "true" process- and observation noise parameters. In other words, optimising these hyperparameters via a grid search is not dissimilar to letting competing methods (e.g., Kalman or particle filtering-type methods) learn the process- and observation noise parameters from other data sets. And if we did the latter, they would not suffer from the misspecification of $Q_{t-1}$ shown in Table 1.

**Clarity**
It would be good if the authors could provide a bit more detail and intuition in the second half of Section 3.1 and in Section 3.2, e.g., about
* the role of the matrices $B_t$ and $C_t$ and why we can be sure that these are well defined.
* the learning-rate matrix (maybe just state the plain gradient-descent update equation which includes this matrix).
There are also a few minor typos, e.g. the notation $\mathbb{E}[p(x_t|y_{1:t-1})]$ does not make sense (presumably $\mathbb{E}[x_t|y_{1:t-1}]$ is meant)? Finally, in Section 4, I'm not sure Equation 13 and the discussion about the delta-method is really needed.

**Questions:**

Some of the "true" model parameters (especially the process and observation noise) in the examples from Sections 5.1 and 5.2 seem to be slightly different than those used in the references cited for these applications. Is there a particular reason for this change? And do the results in these sections differ qualitatively if the parameters from the literature were used?

I am willing to raise my score if the authors can show that the results in Sections 5.1--5.2 are robust to the choice of model parameters.

---

> ### Author Response · Authors · 2023-11-11
>
> Thank you for your constructive review and honest assessment of our work! We are preparing a revised manuscript that will hopefully address your concerns. Please see our initial responses below.
>
> [True model parameters]
>
> - The Lorenz Attractor system was specified such that we could show some level of robustness to model parameters. The Lorenz Attractor follows the most classical setup for the differential equation parameters (sigma = 10, beta = 8/3, and rho = 28), which is known to induce chaotic behavior. We chose the initial starting distribution such that we can sample a different behavior with every starting seed (see Appendix A.5.2). By ensuring that every seed essentially produces a different filtering problem, we can get a good measure of efficacy across 100 starting initializations.
> - We selected the Lorenz system and the current Lorenz configuration to show that the method is robust to many different filtering problems! We found that other configurations (in literature) were less “chaotic” and varied paper to paper with no established standard.
>
> [Choice of hyperparameters]
>
> - The UKF, PF, and EKF have access to the true filtering equations in the toy nonlinear system, so they cannot be improved by a grid search unless there is an improper assumption (misspecified Q). The implicit MAP filter must always perform a grid search, hence why it is not sensitive to a “user mistake” of providing the wrong Q. “User mistake” versus “model misspecification” is a subtle but important distinction that we will fix during revisions. It is more accurate, for experiment 1, to say that the “implicit MAP filter is not sensitive to misspecification by the practitioner because it always requires a grid search” rather than “there is no sensitivity to misspecification”.
> - Your point about choice of hyperparameters applies more to experiment 2, where the provided process noise Q is not necessarily accurate, so a grid search should be done for the baseline methods. Numerical integration introduces additional noise to the process, so methods should ideally search for a process noise covariance Q that accounts for numerical integration errors (Q = provided Q + numerical integration errors). In revisions, we will introduce additional results for the Lorenz experiment where we perform this grid search for the EKF and UKF. In doing so, we can improve both of these methods and address your point. We show that the “best” the EKF and UKF can do still does not beat the proposed method.
>
> [Clarity]
>
> - We will revise 3.1 and 3.2 to include your suggestions and fix minor typos.
>
> [A proposed fix for manuscript revision to show choice robustness]
>
> - We are happy to delegate the current table for Experiment 1 to the Appendix and replace it with results where the true process noise Q = 1, 3, and 5 (instead of showing only Q = 3). This would show the reader performance for each baseline and the proposed method in low, medium, and high process noise cases. As stated above, the Lorenz system already naturally emits many different filtering problems (addressing Section 5.2) and this change would address robustness to model parameters in the 5.1 system. Would this revision increase your assessment of the paper?
>
> Thank you again for your feedback, especially the point about robustness to true model parameters. We are adjusting 5.1 to address this and we believe 5.2 already accomplishes this. We look forward to discussion!

---

> ### Author Response · Authors · 2023-11-17
> **Response: New version of the paper uploaded**
>
> Thank you for the questions and weaknesses you brought up in your review. We just posted a revised version that hopefully addresses all of your concerns and increases your assessment of our paper. Please see the top-level comment for a list of changes to the paper.
>
> [Response to weaknesses/questions]
> - We address all of your concerns in the new versions. Most importantly, the newest version tests 9 systems for the toy nonlinear system instead of 1 and 4 systems for the stochastic lorenz attractor instead of 1. These experiments cover the common range used in literature for running these systems and should address your robustness concern. Results remain consistent.
>
> Please let us know if you have any questions or further concerns. It would be great to hear what you think about these adaptations.

---

### Official Review · Reviewer_Y38s · 2023-11-06

**Soundness:** 3 good
**Presentation:** 3 good
**Contribution:** 3 good
**Rating:** 6
**Confidence:** 4

**Summary:**

This paper proposes to do inference in non-linear, non-Gaussian state space models using an introduced technique termed the "implicit MAP filter". The key idea underlying the proposed method is to build on an observation of Santos (1996) that connects regularized least squares solutions to those obtained by running truncated gradient descent. Translated to the state space model inference context, the choice of steps implicitly defines a prior on the transition process, hence truncated optimization can be viewed as doing Bayesian MAP estimation in the state space model. The authors demonstrate their proposed method on two classical examples from the Particle Filter literature and an online inference task utilizing a pretrained network. It is shown that the proposed method is competitive with standard techniques, and works particularly well on the Lorenz-96 model.

**Strengths:**

The question that the paper attempts to address, namely online learning of state space models, is important in a variety of contexts and the authors offer a concrete pathway to do so, via their proposed optimization framework. The paper provides a potential novel direction that can be followed and certainly developed further, due to, notably, the computational cost savings from looking at optimization as an alternative to (explicit) filtering approaches. The paper is clearly written and easy to read, and the results can be reproduced without much difficulty.

**Weaknesses:**

The interpretation of the number of steps as a prior (or as a regularizer) is not very intuitive. The authors should provide at least some minimal guidance on tuning the number of optimizer steps K and selecting the optimizer itself.

The authors should note some of the modern KF based research in this space (Chang et al. 2023 (https://arxiv.org/abs/2305.19535), Titsias et al. 2023 (https://arxiv.org/abs/2306.08448)) that successfully works on larger-scale problems than the ones considered in this paper.

The experiments chosen are somewhat small-scale given that the authors' argument for not having access to estimates of uncertainty is having a scalable method.

Since the authors explicitly connect the choice of optimizer to defining a prior, they should for example compare different optimizers to one another on the same problem. Have you tried doing this?

**Questions:**

My questions mainly concern the practical uses and developing a better understanding of the proposed method.

Table 1 states that the MAP methods are not dependent on the process noise covariance Q_t-1, which is true, but they are dependent on the number of steps K, which as stated earlier, in e.g., 3.3, implicitly defines a prior. The authors however state that there is no sensitivity to missspecification (last paragraph of Page 6). This should be clarified.

How robust is the method to the number of steps K? What if you run the optimizer essentially until convergence every time, truncating many steps ahead? What if you only do one step of the optimizer. Perhaps the method is not too sensitive to this, but this should be verified and made clear and this aspect of the method should be studied further.

How would the method compare to some of the more modern KF-based methods mentioned earlier, especially on the neural network problem and with some of the datasets that are used there?

In practical contexts, we often don't have access to a pretrained model. How would the method perform when the model is trained completely from scratch?

---

> ### Author Response · Authors · 2023-11-11
>
> Thank you for your constructive review and honest assessment of our work! We are preparing a revised manuscript that will hopefully address your concerns. Please see our initial responses below.
>
> [Comparing different optimizers and robustness to the number of steps K]
>
> - We address your concerns about modifying the number of steps K in Tables 5, 7 in the Appendix. The optimizers are not robust to large changes in K, since this partially encodes the balance between process noise and measurement noise! We also compare different optimizers in Tables 4, 6 in the Appendix.
> - We are hoping that Tables 4-7 include all of the desired information and fully address your concerns!
>
> [Number of steps as prior and selecting the optimizer]
>
> - We agree that more clarity is required with regards to intuition. We are revising section 4.2 to address this.
> - We detail the exact procedure for tuning the number of optimizer steps K and selecting the optimizer itself in Appendix A.4 but please let us know if we can be more clear.
>
> [scale of experiments]
>
> - Since we are using optimizers that are deployed at massive scale for DNNs, we focused on verifying that the proposed method can operate as a Bayesian filter.
> - Since we are comparing to classical Bayesian filters that do not scale like the proposed method, the choice of neural network for the yearbook dataset cannot be too large.
> - We hope these two points clarify the choice of scale.
>
> [implicit MAP without a pre-trained model]
>
> - You are right that pre-trained models are not always available. However, without a pre-trained model, the adaptation would be relatively simple. In an explicit Bayesian filter (ie EKF), one would assume a very large prior covariance if there is no starting information. This is equivalent to, in our implicit formulation, optimization to convergence (because the measurement noise << prior, so you just fit the likelihood). We opted against this style of running this experiment because there would be difficulties in running the particle filter baseline if the transition distribution is not already centered on a relatively good parameterization.
> - We will clarify in Section 5.3 that we can run this implicit MAP filter easily without a pre-trained model by fitting the likelihood exactly at the first timestep.
>
> [Model misspecification]
>
> - We agree that we can clarify “misspecification” for sections 5.1 and 5.2. We will make 5.1 more sound and add results to 5.2 that clarify our meaning of “misspecification”.
>
> [modern KF based research]
>
> - Thank you for referencing two literature sources that should be mentioned in the related works. Similar to the Auvinen et al. (2010) and Bardsley et al. (2013) papers (mentioned in related works of current version), Chang et al. (2023) constructs a low-rank EKF. However, these low-rank filtering methods are not straightforward to implement (relative to proposed method) and are sensitive to the convexity of the loss landscape (most deep learning libraries have all the optimizers we tested and an implicit MAP filter can be implemented in minutes). We believe that Chang et al. (2023) is outside the scope of extremely straightforward large-scale filtering.
> - In the yearbook example, our objective was to test an explicit approach (VKF) and our implicit approach on equal standing of simplicity.
> - Titsias et al. (2023) focuses on dealing with the non-stationarity of the convex readout layer in a continual learning setting, using a simple formulation of a Kalman filter, and focusing more on the detection of transitions. This puts Titsias et al. (2023) slightly outside the scope of our paper because it is not explicitly designed for evolving every weight of a pre-trained network.
>
> Thank you again for the feedback, especially the references to Chang et al. (2023) and Titsias et al. (2023). We are happy to discuss any of these points!

---

> ### Author Response · Authors · 2023-11-17
> **Response: New version of the paper uploaded**
>
> Thank you for the questions and weaknesses you brought up in your review. We just posted a revised version that hopefully addresses all of your concerns and increases your assessment of our paper. Please see the top-level comment for a list of changes to the paper.
>
> [Response to weaknesses/questions]
> - We believe that paper now fully addresses the following concerns: 1) comparing different optimizers and robustness to the number of steps K, 2) number of steps as prior and selecting the optimizer, 3) implicit MAP without a pre-trained model, and 4) model misspecification. We cite Chang et al. (2023) in the related works but do not include an experiment. Our justification is that we are proposing a very simple general filtering method that happens to scale rather than a method for making existing filters scale to larger state spaces. Titsias et al. (2023) focuses on a kind of change-point detection using a standard Kalman filter. It certainly can be used during follow up work (to determine the process noise in real time), but we do not cite it in this work.
>
> Please let us know if you have any questions or further concerns. It would be great to hear what you think about these adaptations.

---

> > ### Comment · Reviewer_Y38s · 2023-11-22
> > **Response**
> >
> > Thank you for your response, I have adjusted my score.

---

### Author Response · Authors · 2023-11-17
**Revised version of the paper just uploaded**

We just uploaded a revised version of the paper. We thank all three reviewers for their excellent feedback. Detailed below are the list of changes.

TL;DR: Added experiments to toy nonlinear system and Lorenz attractor that significantly improve argument. Rewrote Section on implicit MAP filter update step to give better intuition. Rewrote the experimental section to be more succinct, precise, and sound with our argument. Added reference to (Chang et al., 2023) in the related work. Fixed typos, improved writing and clarity throughout the paper.

[Complete List of Changes]
- Stochastic Lorenz Attractor results: We added results where we specified a grid search for the EKF and UKF instead of only using the true model parameters. We make a stronger “misspecification argument” here and the experiments are more sound given this grid search for the baseline methods. We ran all methods for three additional process noise specifications. In total, we have results for alpha = 1, 5, 10, 20. We report alpha = 10 in the main section (as before) and alpha = 1, 5, 20 in the Appendix. Optimized EKF and Implicit MAP Filter were the best and had statistically equivalent performance. However, Implicit MAP Filter has several distinct advantages we mention in the paper. We believe that 5.2 is much more sound and the advantages of our approach are much more clear now.
- Toy nonlinear system results: the optimizer misspecification argument we originally included is now being made in Section 5.2. In the place of the original 5.1 results, we show results over 9 different true model parameters (originally we just showed results for one set of true model parameters). The results are consistent and show that the proposed method can be competitive with the UKF, PF as before…except now it is over 9 toy nonlinear systems instead of 1. We believe this makes the paper significantly stronger.
- Added 5 more tables to the Appendix showcases additional results and added additional writing to the extended experimental section. The reader now has access to a much richer set of results.
- Yearbook dataset results: We made the section more succinct and added a sentence about how this approach is easily adapted to the case where we do not start with a pre-trained network.
- Implicit MAP filter update rule section: completely re-written to offer better intuition for what is happening at the update step and how the choice of optimizer, number of steps, and initialization implicitly chooses the prior.
- Removed unnecessary delta-rule explanation when introducing the implicit MAP filter
- Added reference to (Chang et al., 2023) in the related work.
- Figure 1 (figure and caption) adapted for better clarity.
- Improved writing and clarity concerns in Bayesian Filtering as Optimization section

---

### Meta-Review · Area_Chair_sRgv · 2023-12-07

**Metareview:**

This article proposes a novel approach to finding the posterior mode in high-dimensional state-space models. Building on the work of Santos (1996), the authors demonstrate that the Bayesian filtering problem can be reformulated as an optimization problem with a time-varying objective function. All reviewers agreed that this paper is clearly written and offers a novel approach that should be of interest. Some questions were raised regarding the scale of experiments, the setting of hyperparameters, sensitivity to misspecification, and the discussion of related approaches. These points were satisfactorily addressed in the authors' responses and the revised version, and I recommend acceptance.

**Justification For Why Not Higher Score:**

As the paper builds heavily on Santos (1996), I don't consider that it would be novel enough for a spotlight/oral.

**Justification For Why Not Lower Score:**

All reviewers agreed that this paper is clearly written and offers a novel approach that should be of interest. Some questions were raised regarding the scale of experiments, the setting of hyperparameters, sensitivity to misspecification, and the discussion of related approaches. These points were satisfactorily addressed in the authors' responses and the revised version

---

### Decision · Program_Chairs · 2024-01-16

Accept (poster)